# MA-BERT: Towards Matrix Arithmetic-only BERT Inference by Eliminating Complex Non-Linear Functions

**Neo Wei Ming[1,2], Zhehui Wang[1], Cheng Liu[3], Rick Siow Mong Goh[1], Tao Luo[1]***

Institute of High Performance Computing (IHPC), Agency for Science, Technology and Research (A*STAR), 1 Fusionopolis Way, 16-16 Connexis, Singapore 138632, Republic of Singapore[1]
School of Computer Science and Engineering, Nanyang Technological University, Singapore[2]
Institute of Computing Technology, Chinese Academy of Sciences[3]
`{neo.weiming, leto.luo}@gmail.com`
`https://github.com/W6WM9M/MA-BERT`

## Abstract

Due to their superior results, Transformer-based models such as BERT have become de facto standards in many Natural Language Processing (NLP) applications. However, the intensive use of complex non-linear functions within the Transformer architecture impairs its computing efficiency and complicates corresponding accelerator designs, because non-linear functions are generally computation-intensive and require special hardware support. In light of this, we propose MA-BERT, which allows matrix arithmetic-only operations in Transformer-based NLP models and achieves efficient inference with negligible accuracy loss. Specifically, we propose four correlated techniques that include approximating softmax with a two-layer neural network, replacing GELU with ReLU, fusing normalization layers with adjacent linear layers, and leveraging knowledge transfer from baseline models. Through these techniques, we are able to eliminate the major non-linear functions in Transformer-based models and obtain MA-BERT with only matrix arithmetic and trivial ReLU operations without compromising on accuracy. With mainly regular matrix arithmetic operations, MA-BERT enables hardware-friendly processing on various computing engines, including CPUs and GPUs. Our experimental results show that MA-BERT achieves up to 27% and 41% reduction in inference time on CPU and GPU, respectively, with comparable accuracy on many downstream tasks compared to the baseline BERT models.

## 1 Introduction

Recently, pretrained Transformer-based models such as GPT (Radford et al., 2018) and BERT (Devlin et al., 2018), have consistently dominated the leaderboards for a variety of NLP tasks and even surpassed the human baseline. Consequently, there has been a strong push for these models to be used in many NLP applications. At the same time, they have become a popular research topic in academia and industry. This in turn led to the creation of even better models such as GPT-3 (Brown et al., 2020), RoBERTa (Liu et al., 2019), and DeBERTa (He et al., 2020), which further stretched the limits of what such models can achieve. Clearly, the advent of Transformer-based models has brought about a paradigm shift in the NLP field from the pre-Transformer era when recurrent neural networks and their variants used to dominate.

Nonetheless, the exceptional performance of these Transformer-based models partly stems from their deep structure which involves a huge number of parameters and operations during a single forward propagation. These characteristics make it challenging to meet timing requirements and implement them on devices with limited memory and computational power. Consequently, many notable works have been proposed to improve their inference performance. In particular to BERT,

---
*Corresponding author

some works seek to compress the model by reducing the number of encoder layers via knowledge distillation (Sanh et al., 2019; Aguilar et al., 2020; Sun et al., 2020; 2019; Jiao et al., 2019; Xu et al., 2020) with a minor accuracy penalty. Other works focus on model pruning (Gao et al., 2021; Gordon et al., 2020; Voita et al., 2019; Wang et al., 2021), which involves eliminating unessential weights or attention heads, leading to sparser models with a reduced number of parameters and computing operations. Additionally, several works leverage quantization (Kim et al., 2021; Bhandare et al., 2019; Zafrir et al., 2019) to reduce the precision of weights and computing operations, which lowers memory demands and speeds up inference.

Despite these works, one non-negligible overhead in Transformer-based models is often overlooked – the intensive use of complex non-linear functions. In BERT, these non-linear functions include the softmax operation, Gaussian Error Linear Unit (GELU) activation function (Hendrycks & Gimpel, 2016), and Layer Normalization (LayerNorm) (Ba et al., 2016). Although these functions undeniably play a role in helping the model to learn better during training time, they become a considerable bottleneck during inference as they are not straightforward to evaluate. Furthermore, in hardware accelerator designs, these non-linear functions typically require separate hardware for acceleration (Liu et al., 2021; Khan et al., 2021), which makes them challenging to be deployed on resource-constrained computing platforms. In Figure 1 and Table 1, we show the breakdown of the cycle budget for the BERT$_{base}$ model. For non-linear operations, the cycle budget is defined as the equivalent number of cycles that the matrix multiply takes to process, given the matrix multiply dimensions and the number of multiplications per cycle (Khan et al., 2021). From the figure, we can see that those non-linear operations can take as much as 43% of the processing time. The computation efficiency of BERT has a huge potential to improve if we can optimize these non-linear operations.

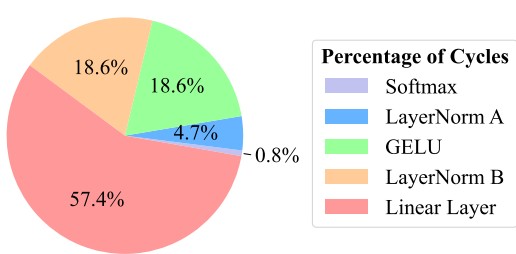

Figure 1: Breakdown of cycle budget

Table 1: Cycle budget for BERT$_{base}$ with sequence length of 128 and 2048 multiplications per cycle (Khan et al., 2021)

| Functions | Cycle Budget | Percentage |
|---|---|---|
| Softmax | 512* | 0.8% |
| LayerNorm A | 36,864 | 4.7% |
| GELU | 147,456 | 18.6% |
| LayerNorm B | 147,456 | 18.6% |
| Linear Layer | 454,656 | 57.4% |

* Per Attention Head

In view of this, we introduce an efficient BERT, termed MA-BERT, which makes a novel attempt to completely eliminate the complex non-linear functions in BERT by substituting them with simpler functions. In MA-BERT, four correlated techniques are applied: (1) approximating softmax with a two-layer neural network, (2) replacing GELU with ReLU, (3) fusing normalization layers with adjacent linear layers, and (4) leveraging knowledge transfer from pretrained baseline models. Through these techniques, MA-BERT achieves matrix arithmetic-only and trivial ReLU operations with negligible accuracy loss.

Our experiments show that the performance of MA-BERT on downstream tasks is on par with the corresponding BERT baseline and yet achieves up to 27% and 41% performance improvement on general CPUs and GPUs, respectively. By eliminating the complex non-linear functions, the majority of the operations in MA-BERT becomes regular matrix-matrix arithmetic and can potentially be deployed on various computing engines, including CPUs and GPUs, without any hardware modification.

## 2 RELATED WORK

While so many works focus on model compression techniques, only some paid attention to the optimization of non-linear functions in BERT and other variants of Transformer-based models. For instance, in MobileBERT, Sun et al. (2020) proposed a thin version of BERT$_{large}$ with reduced hidden dimensions. As part of their operational optimization, they replaced GELU with ReLU and

removed LayerNorm to further reduce MobileBERT's latency. However, their results showed that the inclusion of these optimizations degrades the model's performance. Kim et al. (2021), on the other hand, quantized the non-linear functions in BERT to 32-bit integer precision, allowing these non-linear functions to be computed using faster integer arithmetic.

Specific to tackling the latency overhead of LayerNorm, Zhang & Sennrich (2019) proposed root mean square layer normalization which reduces the amount of computation compared to LayerNorm by only using root mean square statistics to normalize inputs. In the field of computer vision, Yao et al. (2021) reintroduced Batch Normalization (BatchNorm) (Ioffe & Szegedy, 2015) as a replacement for LayerNorm (Ba et al., 2016) into vision Transformers and demonstrated that the resulting architecture performs on par with its LayerNorm-based counterpart. Similarly, Shen et al. (2020) proposed PowerNorm as a replacement for LayerNorm in Transformers and demonstrated that PowerNorm can outperform LayerNorm in neural machine translation and language modelling tasks. Due to its proven performance and use of batch statistics, we have decided to adopt PowerNorm in our paper to minimize the overhead introduced by the normalization operations in BERT during inference.

Leaning onto the hardware side, Yu et al. (2022) proposed a general approximation framework that enabled them to convert all the non-linear operations into table look-ups. This helped to reduce the cost of non-linear functions but resulted in new types of operations and corresponding hardware.

Different from these works, our work strives to completely eliminate the complex non-linear functions in BERT and achieve matrix arithmetic-only operations with trivial ReLU, which could benefit inference on both general computing units and accelerator designs for edge applications.

## 3 METHOD

This section gives a brief overview of the non-linear functions that exist in BERT and highlights their computation inefficiency during inference. Following that, more efficient methods are proposed.

### 3.1 SOFTMAX: APPROXIMATING WITH A 2-LAYER NEURAL NETWORK

Every encoder layer in BERT contains a multi-head self-attention (MHA) sublayer followed by a feed forward network (FFN) sublayer. The MHA sublayer endows the model with the ability to capture various dependencies between different parts of an input sequence (Vaswani et al., 2017) and is defined as follows:

$$\text{MultiHead}(X) = \text{Concat}(\text{head}_1, ..., \text{head}_h)W^O \tag{1}$$

$$\text{where head}_i = \text{softmax}(\frac{XW_{Q_i}W_{K_i}^T X^T}{\sqrt{d}})XW_{V_i} \tag{2}$$

In each self-attention head, a $s \times s$ scaled dot product of the query and key matrices is passed through a softmax function to obtain the weights on the value matrix. Here, $s$ denotes the sequence length. The softmax function is computed as:

$$\text{softmax}(x_i) = \frac{\exp(x_i)}{\sum_{j=1}^{s} \exp(x_j)} \tag{3}$$

From Eq. 3 it is clear that all inputs along the sequence dimension have to be processed before the softmax value of a particular input can be computed. This becomes prohibitively expensive as the input sequence length increases (Du et al., 2019; Stevens et al., 2021). Moreover, the softmax function contains expensive exponentiation and division operations which further complicate accelerator designs (Hu et al., 2018). In view of this, we decided to find an alternative for the softmax function.

The universal approximation theorem (Hornik et al., 1989) states that a neural network with a single hidden layer and non-linear activation function can approximate any function, given sufficient hidden neurons. As such, we propose to substitute the softmax function in each layer of BERT with a simple 2-layer neural network. ReLU is chosen as the activation function of the hidden layer with hardware reuse in mind since we will also be adopting it in the FFN sublayer. Note that this modification will inevitably fix the maximum input sequence length of our modified model to the input layer size of the neural network. To eliminate the effects of any padding tokens in an input sequence,

we multiplied the scaled dot product of the query and key matrices with the input attention mask before feeding it as an input to the neural network.

Given that the same softmax function is being approximated, it should ideally be possible to approximate softmax using the same neural network across the encoder layers, allowing us to reduce the parameters introduced.

## 3.2 GELU: Replacing with ReLU

Unlike the original Transformer architecture proposed by Vaswani et al. (2017), many Transformer-based NLP models such as BERT use the GELU activation function instead of ReLU in their FFN layers to enhance their learning capability. GELU is defined as follows:

$$\text{GELU}(x) = x \cdot \Phi(x), \text{ where } \Phi(x) = \frac{1 + \frac{2}{\sqrt{\pi}} \int_0^{\frac{x}{\sqrt{2}}} \exp(-t^2) \, dt}{2} \tag{4}$$

According to Eq. 4, it is clear that GELU involves complex operations such as exponentiation and integration which are computation-intensive and hard to be implemented efficiently in accelerator designs. One possible way is to employ a lookup table to simplify the operations, but doing so entails a trade-off between accuracy and hardware utilization. Other implementations involving function approximations (Hendrycks & Gimpel, 2016; Kim et al., 2021) also exist, but they often involve multiple operations which end up consuming a notable portion of inference time (Pati et al., 2021). In contrast, ReLU simply outputs zero or the input value depending on the input's sign and can be implemented easily in hardware using a single multiplexer. With the potential simplification in hardware and speed-up during inference (Sun et al., 2020), we propose to bring ReLU back as a replacement for GELU.

## 3.3 LayerNorm: Fusing Normalization Layers with Adjacent Linear Layers

Every encoder layer in BERT involves two LayerNorm sublayers – one after the MHA sublayer and another after the FFN sublayer. LayerNorm involves the calculation of the mean $\mu$ and standard deviation $\sigma$ statistics along the feature dimension of size $K$, followed by an affine transformation:

$$\bar{x}_i = \gamma \cdot \frac{x_i - \mu}{\sigma} + \beta, \text{ where } \mu = \frac{1}{K} \sum_{i=1}^{K} x_i, \ \sigma = \sqrt{\frac{1}{K} \sum_{i=1}^{K} (x_i - \mu)^2} \tag{5}$$

This remains true even during inference and demands the need to wait for all features of an input token to be processed before these statistics can be computed. It would be ideal if some form of batch statistics can be used as an alternative to normalize the inputs. Doing so will eliminate the need to compute these statistics and present us with the opportunity to fuse with adjacent linear layers in BERT during inference.

Recently, Shen et al. (2020) proposed PowerNorm as a replacement for LayerNorm in Transformers and demonstrated that PowerNorm can outperform LayerNorm in neural machine translation and language modelling tasks. Similar to BatchNorm, PowerNorm normalizes the inputs along the batch dimension of size $B$ and uses running statistics during inference. However, it enforces only unit quadratic mean:

$$\bar{x}_i = \gamma \cdot \frac{x_i}{\psi} + \beta, \text{ where } \psi^2 = \frac{1}{B} \sum_{i=1}^{B} x_i^2 \tag{6}$$

During inference, the parameters $\gamma$, $\psi$, and $\beta$ become constants that can be merged with the parameters in the adjacent linear layers. Specifically, if we were to replace LayerNorm with PowerNorm, the PowerNorm after the MHA sublayer can be fused with either the linear projection after the concatenation operation in Eq. 1 or the linear projection from the input layer to the intermediate layer of the subsequent FFN. Similarly, the PowerNorm after the FFN sublayer can be fused with either the linear projection from the intermediate layer to the output layer of the FFN or with the computation of the key, query, and value matrices in the next encoder layer. That said, due to the post-normalization architecture of BERT, it is not possible to do a complete fusion. Nonetheless, it is still possible to reduce the two PowerNorm operations in every MA-BERT encoder layer to a single scaling operation during inference, which offers us a considerable reduction in computation.

Given the prospect of minimizing the overhead of the normalization operation during inference, we propose to substitute LayerNorm in BERT with PowerNorm. Note that, unlike their work, we did not include a layer-scale operation before PowerNorm to prevent any additional overhead from being introduced.

### 3.4 KNOWLEDGE TRANSFER

With the three optimization techniques proposed in the previous subsections, we successfully achieve a BERT model with only matrix arithmetic and trivial ReLU operations. However, as a result of the modifications, a large accuracy gap exists between the untrained MA-BERT and the baseline. In order to bridge this gap, we propose to leverage knowledge transfer.

Many studies have demonstrated the effectiveness of knowledge distillation (Hinton et al., 2015) in bridging the performance gap between a student model and a teacher model. The student model is trained to mimic the behaviour of a teacher model using a well-designed cost function that takes into account the differences between the student output and the teacher output. Unlike many studies which used knowledge distillation to compress BERT into smaller models, we leverage knowledge transfer with the intention to train a model of the same depth and hidden dimensions with a more efficient and hardware-friendly architecture. Note that we are using the term knowledge transfer rather than knowledge distillation, given that the latter is often associated with a smaller model learning from a larger model. The following describes our procedure to transfer knowledge from an existing pretrained BERT$_{base}$ model to our modified model, MA-BERT$_{base}$.

#### 3.4.1 TEACHER MODEL

We used the pretrained BERT$_{base}$ model from HuggingFace as our teacher model. All parameters of the teacher model were frozen to prevent the student model from chasing a moving target.

#### 3.4.2 STUDENT MODEL

For the student model (MA-BERT$_{base}$), we aggregated and implemented the three modifications described in the previous subsections on a BERT$_{base}$ model and initialized it with its pretrained parameters, wherever possible. Furthermore, the newly added 2-layer neural network was pretrained in advance to approximate the softmax function. For simplicity, the hidden layer size of the 2-layer neural network was set to be the same as the input sequence length. The proposed modifications were only applied on the embedding and encoder layers of BERT, and all parameters in these layers were left unfrozen, except for the position and token type embeddings.

#### 3.4.3 KNOWLEDGE TRANSFER OBJECTIVE

In order for the student to replicate the teacher's behaviour as closely as possible, we adopted a similar layer-wise knowledge transfer technique proposed by Jiao et al. (2019).

First of all, we used the cross entropy (CE) between the student's logits $\mathbf{z}^S$ and the actual labels $\mathbf{y}$ to penalize any wrong classification:

$$L_{hard} = CE(\mathbf{z}^S, \mathbf{y}) \tag{7}$$

Next, we considered the cross entropy between the student's logits and the teacher's logits $\mathbf{z}^T$:

$$L_{soft} = CE(\mathbf{z}^S/t, \mathbf{z}^T/t) \tag{8}$$

Compared to the actual labels, the teacher's logits provide much more information for the student to learn. A higher temperature $t$ smoothens the probability distribution, which allows the student to better learn the intricacies of the teacher model.

Thirdly, to ensure each layer of our student model, including the embedding layer, learns from the corresponding layer of the teacher model, we included the following mean-squared error (MSE) objective:

$$L_{hidn} = \sum_{i=0}^{l} MSE(\mathbf{H}_i^S, \mathbf{H}_i^T) \tag{9}$$

$\mathbf{H}_i^S$ and $\mathbf{H}_i^T$ are the hidden states of the student and teacher models, respectively, and $l$ is the number of layers in the model.

Lastly, given that we are replacing the softmax function in the MHA sublayer, we also penalized the discrepancy between the student's attention outputs $\mathbf{A}_i^S$ and the teacher's attention outputs $\mathbf{A}_i^T$ using:

$$L_{attn} = \sum_{i=1}^{l} MSE(\mathbf{A}_i^S, \mathbf{A}_i^T) \tag{10}$$

Altogether, the overall objective function for knowledge transfer can be summarized as follows:

$$L_{net} = (1 - \alpha) * L_{hard} + \alpha * L_{soft} + \beta\left(L_{hidn} + \gamma * L_{attn}\right) \tag{11}$$

$\alpha$ is used to balance the importance of achieving the actual hard target and the teacher's soft target, $\beta$ is used to balance the importance of mimicking the teacher's internal layer outputs relative to mimicking the final output, and $\gamma$ is a factor to scale $L_{attn}$ to a similar magnitude as $L_{hidn}$. With this overall objective, the student is compelled to not only mimic the final output of the teacher but also its hidden layer and attention outputs. Throughout our study, we kept $t = 15$, $\alpha = 0.9$, $\beta = 1$, and $\gamma = 100$.

## 4 EXPERIMENTS

In this section, we detail our pretraining setup and evaluate the performance of MA-BERT$_{base}$ on several downstream tasks. We also describe the details of additional studies that we have conducted that attempt to highlight and exploit the benefits of our approach. Throughout this section, all results pertain to a model input sequence length of 128 tokens.

### 4.1 PRETRAINING SETUP

We pretrained MA-BERT$_{base}$ via knowledge transfer on a concatenation of BookCorpus (Zhu et al., 2015) and English Wikipedia using only the Masked Language Modelling objective (Devlin et al., 2018) with a masking probability of 15%. We used AdamW optimizer with a learning rate of $2e^{-5}$, 10,000 warm-up steps from a base learning rate of $5e^{-7}$, and a linear decay of learning rate. The training was done with a batch size of 256 over 3 epochs, with each epoch using a differently masked dataset, on a single NVIDIA GeForce RTX 3090 for a total of 3.5 days.

### 4.2 EVALUATION ON DOWNSTREAM TASKS

To evaluate the performance of MA-BERT$_{base}$, we used the GLUE benchmark (Wang et al., 2018), which consists of nine natural language understanding tasks, and the IMDb sentiment classification task (Maas et al., 2011). Following many studies, we excluded WNLI given its small dataset. For each task, we applied task-specific knowledge transfer by having MA-BERT$_{base}$ learn from a fine-tuned BERT$_{base}$. The same knowledge transfer objective and hyperparameter settings outlined in section 3.4.3 were used. As our goal is to replicate the performance of the teacher, our baseline for comparison is simply the teacher, which is BERT$_{base}$ in this case.

For smaller datasets (CoLA, MRPC, RTE, and STS-B), we used a batch size of 16 and fine-tuned for 10 epochs, and for the remaining datasets, we used a batch size of 32 and fine-tuned for 5 epochs. The same learning rate of $2e^{-5}$ was used across all tasks and the sequence length was fixed at 128. To account for the variance of different runs, we report the median of 5 runs with different random initialization seeds for each task. The results are shown in Table 2 and 3. Note that the teacher with the best performance among the 5 runs was used to finetune the student during task-specific knowledge transfer. The best performance of both the teacher and student is reported under Appendix A.1.

On average, despite the modifications, MA-BERT$_{base}$ was able to achieve similar performance on the downstream tasks as compared to BERT$_{base}$. In fact, for several tasks, MA-BERT$_{base}$ managed to outperform its teacher. This demonstrates that adopting our approach does not lead to a degradation in the average performance of the model.

## 4.3 Hyperparameter Tuning of $\alpha$ and $\beta$

Prior to the collection of results in Table 2 and 3, we conducted a simple hyperparameter study to assess the impact of our choices for $\alpha$ and $\beta$ in our knowledge transfer objective during task-specific knowledge transfer. We varied $\alpha$ and $\beta$, which effectively control how strictly the student should replicate the teacher's final and hidden layer outputs, and evaluated MA-BERT$_{base}$ on the 4 smaller datasets of the GLUE benchmark. The results are shown in Table 4. Empirically, we found that MA-BERT$_{base}$ performs the best on downstream tasks under the strict guidance of the finetuned BERT$_{base}$ model with the optimal hyperparameter values of $\alpha = 0.9$ and $\beta = 1.0$.

Table 2: Median performance on the development set of GLUE benchmark. Matthews correlation is reported for CoLA while Pearson correlation is reported for STS-B. Accuracy is reported for the remaining tasks. MNLI is the average of matched and mismatched sets. AVG denotes the average of the 8 GLUE tasks.

| Model | CoLA | MNLI | MRPC | QNLI | QQP | RTE | SST-2 | STS-B | AVG |
|---|---|---|---|---|---|---|---|---|---|
| BERT$_{base}$ | 58.8 | 84.5 | 86.3 | 91.4 | 91.2 | 69.3 | 92.8 | 89.3 | 83.0 |
| MA-BERT$_{base}$ | 59.8 | 84.8 | 87.5 | 91.0 | 91.2 | 68.6 | 92.5 | 87.8 | 82.9 |
|  | (▲1.0) | (▲0.3) | (▲1.2) | (▼0.4) | (0.0) | (▼0.7) | (▼0.3) | (▼1.5) | (▼0.1) |
| MA-BERT$_{base}$ | 59.0 | 84.7 | 87.0 | 91.1 | 91.1 | 70.0 | 92.4 | 87.3 | 82.8 |
| (Shared Softmax) | (▲0.2) | (▲0.2) | (▲0.7) | (▼0.3) | (▼0.1) | (▲0.7) | (▼0.4) | (▼2.0) | (▼0.2) |
| DistilBERT | 55.1 | 82.3 | 85.0 | 88.4 | 90.3 | 62.5 | 90.8 | 87.1 | 80.2 |
| MA-DistilBERT | 51.3 | 82.3 | 86.7 | 88.4 | 90.3 | 65.3 | 90.8 | 87.5 | 80.3 |
|  | (▼3.8) | (0.0) | (▲1.7) | (0.0) | (0.0) | (▲2.8) | (0.0) | (▲0.4) | (▲0.1) |
| DistilRoBERTa | 58.3 | 84.3 | 85.8 | 91.2 | 90.9 | 69.3 | 92.9 | 88.4 | 82.6 |
| MA-DistilRoBERTa | 51.7 | 83.8 | 86.0 | 90.8 | 90.7 | 67.5 | 91.6 | 87.8 | 81.2 |
|  | (▼6.6) | (▼0.5) | (▲0.2) | (▼0.4) | (▼0.2) | (▼1.8) | (▼1.3) | (▼0.6) | (▼1.4) |
| MA-DistilRoBERTa | 47.6 | 83.2 | 83.8 | 90.5 | 90.5 | 65.7 | 90.8 | 87.1 | 79.9 |
| (Linear Attention) | (▼10.7) | (▼1.1) | (▼2.0) | (▼0.7) | (▼0.4) | (▼3.6) | (▼2.1) | (▼1.3) | (▼2.7) |

Table 3: Accuracy on IMDb test set.

| Model | IMDb |
|---|---|
| BERT$_{base}$ | 89.3 |
| MA-BERT$_{base}$ | 89.6 (▲0.3) |
| MA-BERT$_{base}$ | 89.6 (▲0.3) |
| (Shared Softmax) | |
| DistilBERT | 88.1 |
| MA-DistilBERT | 88.2 (▲0.1) |
| DistilRoBERTa | 89.9 |
| MA-DistilRoBERTa | 89.8 (▼0.1) |
| MA-DistilRoBERTa | 89.6 (▼0.3) |
| (Linear Attention) | |

Table 4: $\alpha$ and $\beta$ hyperparameter tuning results on selected GLUE tasks for MA-BERT$_{base}$. AVG denotes the average of the 4 selected GLUE tasks.

| $\alpha, \beta$ | CoLA | MRPC | RTE | STS-B | AVG |
|---|---|---|---|---|---|
| 0.3, 0.3 | 58.3 | 84.3 | 61.0 | 86.6 | 72.6 |
| 0.6, 0.6 | 59.9 | 86.5 | 64.6 | 86.7 | 74.4 |
| 0.9, 0.9 | 60.4 | 87.0 | 68.2 | 87.7 | 75.8 |
| 0.3, 1.0 | 60.0 | 86.8 | 65.0 | 87.1 | 74.7 |
| 0.6, 1.0 | 59.8 | 86.5 | 65.3 | 87.1 | 74.7 |
| 0.9, 1.0 | 59.8 | 87.5 | 68.6 | 87.8 | **75.9** |

## 4.4 Sharing Softmax Approximation Neural Network Across Layers

In the previous subsection, the softmax function in each encoder layer of MA-BERT$_{base}$ was approximated using a different 2-layer neural network. However, given that it is the same softmax function, it would make sense if we can share the same 2-layer neural network across all layers. Doing so would also help us reduce the additional parameters introduced by our approach. In fact, we found that the weights learnt by the 2-layer neural network in each encoder layer in MA-BERT$_{base}$ are largely similar. A visualization of the weights learnt can be found in Figure 5 under Appendix A.2. We attempted to retrain MA-BERT$_{base}$ with parameter sharing in the softmax approximation neural network using the same training procedure. Empirically, we found that sharing the 2-layer neural network to approximate softmax preserves the performance of the modified model.

### 4.5 Versatility Study

To evaluate the versatility of our approach, we also attempted our modifications and training procedure on HuggingFace's DistilBERT (Sanh et al., 2019) and DistilRoBERTa to produce MA-DistilBERT and MA-DistilRoBERTa, respectively. Following DistilRoBERTa's training procedure, MA-DistilRoBERTA was instead pretrained on the OpenWebText (Gokaslan & Cohen, 2019) dataset. The remaining training procedure was kept unchanged. We observed that both MA-DistilBERT and MA-DistilRoBERTa suffered a significant drop in performance on the CoLA task. Nonetheless, in terms of average performance, MA-DistilBERT managed to slightly outperform its teacher, DistilBERT. On the other hand, although MA-DistilRoBERTa was not able to achieve similar average performance as DistilRoBERTa, it was still able to outperform DistilBERT, which has the same hidden dimensions and depth.

This versatility study also illustrates that our approach is orthogonal to model compression via knowledge distillation. We can first distil a model into a shallower version, then apply our modifications to make it even more efficient and hardware friendly for inference.

### 4.6 Adopting Linear Self-Attention

With the use of a neural network to approximate the softmax operation, we have inevitably introduced additional matrix multiplication operations. The increase in matrix multiplication scales in $O(s^3)$ if the hidden layer size of the neural network follows the input sequence length $s$. That said, for relatively short sequence lengths, the increase is marginal. For $s = 128$, the increase in the total number of matrix multiplication compared to the baseline is around 5.4% (Figure 2). Nonetheless, the issue of scalability comes to light.

For longer sequence lengths, we suggest a possible solution that can reduce this overhead. Recently, Wang et al. (2020) proposed self-attention with linear complexity. This is achieved by projecting the key and value matrices from dimensions $\mathbb{R}^{s \times d}$ into smaller dimensions of $\mathbb{R}^{k \times d}$ using the projection matrices $E, F \in \mathbb{R}^{k \times s}$ and setting $k \ll s$:

$$head_i = softmax(\frac{XW_{Q_i}W_{K_i}^T X^T E^T}{\sqrt{d}})FXW_{V_i} \tag{12}$$

Here, $d$ denotes the hidden dimension of the model while $k$ denotes the new projected dimension. By adopting linear self-attention, softmax has to be computed only along the reduced dimension $k$, which enables us to use a smaller 2-layer neural network to replace softmax. This not only helps to reduce the overheads introduced by the additional neural network to approximate softmax but also allows us to leverage the speedup offered by linear self-attention. For instance, by letting $k = 0.5s$, significant savings can be achieved as illustrated in Figure 2. To take this even further, it is also possible to fuse the linear operations in the softmax approximation neural network with both the projection of the key and value matrices during inference, hiding the overhead introduced.

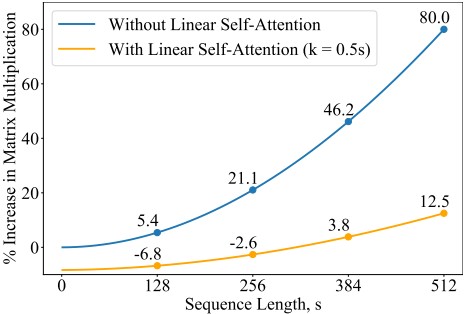

Figure 2: Percentage increase in matrix multiplication in an encoder layer of MA-BERT with and without linear self-attention compared to BERT against input sequence length.

To test this out, we trained MA-DistilRoBERTa with linear attention by setting $k = 64$ using the same procedure. The loss term $L_{attn}$ was removed from the knowledge transfer objective due to the different attention output dimensions of the modified model and DistilRoBERTa. The results

are reported in Table 2 and 3. We noticed a 3.3% drop in the performance of the modified model compared with the baseline DistilRoBERTa. Nonetheless, it was still able to achieve comparable average performance (0.37% drop) as baseline DistilBERT. A revision of the training procedure might be necessary to further improve its performance and we will leave this to future studies.

## 4.7 EVALUATION ON COMPUTATIONAL EFFICIENCY

To evaluate the time efficiency of MA-BERT during inference, we implemented DistilBERT and MA-DistilBERT on two different hardware, namely, i7-1260P CPU and RTX 3090 GPU. We used C++ with Eigen3 library for CPU implementation and Cuda C++ with Thrust and cuBLAS libraries for GPU implementation. Using the IMDb test dataset, we measured the average time taken for a single encoder layer inference. To account for the variance in the time taken, we report the median of 5 different runs. The results are summarized in Table 5. Despite the higher number of matrix multiplications involved, MA-DistilBERT is able to offer an inference time speedup of up to $1.27\times$ on CPU and $1.41\times$ on GPU. Note that the speedups reported also applies to MA-BERT$_{base}$ and MA-DistilRoBERTa, given that their encoder layers are identical.

Table 5: Forward propagation time of a single encoder layer in DistilBERT and MA-DistilBERT on CPU and GPU. All parameters and calculations were in 32-bit floating-point precision.

|  | i7-1260P CPU | RTX 3090 GPU |
|---|---|---|
| DistilBERT | 93.0 ms | 2.37 ms |
| MA-DistilBERT | 68.0 ms | 1.39 ms |
| Speedup | $1.27\times$ | $1.41\times$ |

## 5 ABLATION STUDY

In this last section, we highlight the importance of having the baseline teacher's guidance during MA-BERT's finetuning phase.

## 5.1 REMOVAL OF TEACHER'S GUIDANCE

To demonstrate the importance of task-specific knowledge transfer during finetuning, we repeated our finetuning procedure on the GLUE tasks by replacing the knowledge transfer objective with only the hard target loss for our student model to optimize. The results are reported in Table 6. We found that without any guidance from the teacher, the finetuning process is often unstable with the student model ending up performing worse in all tasks.

Table 6: Effects of removing the teacher's guidance during finetuning on the GLUE tasks.

| Model | Knowledge Transfer | CoLA | MNLI | MRPC | QNLI | QQP | RTE | SST-2 | STS-B |
|---|---|---|---|---|---|---|---|---|---|
| MA-BERT$_{base}$ | ✓ | 59.8 | 84.8 | 87.5 | 91.0 | 91.2 | 68.6 | 92.5 | 87.8 |
|  | ✗ | 31.5 | 35.3 | 80.9 | 50.5 | 63.2 | 61.4 | 86.8 | 81.6 |
| MA-DistilBERT | ✓ | 51.3 | 82.3 | 86.7 | 88.4 | 90.3 | 65.3 | 90.8 | 87.5 |
|  | ✗ | 1.8 | 35.3 | 68.4 | 50.7 | 63.2 | 53.1 | 85.6 | 85.3 |

## 6 CONCLUSION

In this paper, we introduced MA-BERT, an efficient and hardware-friendly version of BERT with all its complex non-linear functions eliminated. Through knowledge transfer during pretraining and finetuning, MA-BERT is able to attain comparable performance to BERT. Yet, compared to BERT, MA-BERT offers a reduction in inference time with potential savings in hardware resources, making it a suitable solution for applications on edge devices. In the future, it would be interesting to study the applicability of our approach to other transformer-based models.

ACKNOWLEDGMENTS

We gratefully acknowledge the support by the Singapore Government's Research, Innovation and Enterprise 2020 Plan (Advanced Manufacturing and Engineering domain) under Grant A1892b0026.

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

# A    APPENDIX

## A.1    PERFORMANCE OF BEST PERFORMING TEACHER AND STUDENT MODELS

During task-specific knowledge transfer, the best-performing finetuned teacher was used as the teacher for the pretrained student model. Table 7 shows the best performance achieved by the baseline teacher models and the modified student models among the 5 runs for each task.

Table 7: Best performance on the dev set of GLUE benchmark and test set of IMDb. Matthews correlation is reported for CoLA while Pearson correlation is reported for STS-B. Accuracy is reported for the remaining tasks. MNLI is the average of matched and mismatched sets

| Model | CoLA | MNLI | MRPC | QNLI | QQP | RTE | SST-2 | STS-B | IMDb |
|---|---|---|---|---|---|---|---|---|---|
| BERT$_{base}$ | 59.4 | 84.9 | 87.5 | 91.5 | 91.4 | 69.7 | 93.0 | 89.5 | 89.5 |
| MA-BERT$_{base}$ | 61.3 | 84.9 | 88.0 | 91.1 | 91.3 | 70.8 | 92.7 | 88.1 | 89.8 |
| | (▲1.9) | (0.0) | (▲0.5) | (▼0.4) | (▼0.1) | (▲1.1) | (▼0.3) | (▼1.4) | (▲0.3) |
| MA-BERT$_{base}$ | 62.1 | 85.0 | 87.3 | 91.3 | 91.2 | 70.4 | 92.8 | 87.6 | 89.7 |
| (Shared Softmax) | (▲2.7) | (▲0.1) | (▼0.2) | (▼0.2) | (▼0.2) | (▲0.7) | (▼0.2) | (▼1.9) | (▲0.2) |
| DistilBERT | 55.5 | 82.5 | 86.0 | 88.7 | 90.4 | 64.3 | 91.4 | 87.4 | 88.2 |
| MA-DistilBERT | 52.4 | 82.4 | 87.0 | 88.7 | 90.4 | 66.4 | 91.4 | 87.7 | 88.3 |
| | (▼3.1) | (▼0.1) | (▲1.0) | (0.0) | (0.0) | (▲2.1) | (0.0) | (▲0.3) | (▲0.1) |
| DistilRoBERTa | 59.8 | 84.5 | 86.3 | 91.3 | 91.0 | 71.8 | 93.0 | 88.8 | 90.2 |
| MA-DistilRoBERTa | 51.8 | 83.9 | 86.8 | 91.1 | 90.8 | 68.2 | 92.0 | 87.9 | 89.8 |
| | (▼8.0) | (▼0.9) | (▲0.5) | (▼0.2) | (▼0.2) | (▼3.6) | (▼1.0) | (▼0.9) | (▼0.4) |
| MA-DistilRoBERTa | 49.7 | 83.4 | 85.8 | 90.6 | 90.5 | 66.4 | 91.1 | 87.5 | 89.7 |
| (Linear Attention) | (▼10.1) | (▼1.1) | (▼0.5) | (▼0.7) | (▼0.5) | (▼5.4) | (▼1.9) | (▼1.3) | (▼0.5) |

## A.2    ANALYSIS OF SOFTMAX APPROXIMATION

After pretraining MA-BERT$_{base}$, we decided to evaluate the effectiveness of the 2-layer neural network in approximating softmax. We fed MA-BERT$_{base}$ with the sample input sequence "the quick brown fox jumps over the lazy dog" and plotted out the multi-head self-attention scores produced in each of its encoder layers. We then compared them with the corresponding encoder layer attention scores in BERT$_{base}$. Figure 3 shows a side-by-side comparison of the attention scores produced in the 4th, 8th, and 12th encoder layers of MA-BERT$_{base}$ and BERT$_{base}$. We observe that the attention scores produced by both models are similar, which demonstrates the effectiveness of the 2-layer neural network in approximating softmax. The same observation is made when we compare the attention outputs between MA-DistilBERT and DistilBERT in Figure 4.

Moreover, we decided to look into the weights that are learnt by the 2-layer neural network to investigate if any meaningful mappings are learnt. Figure 5 shows the learnt weights mapping the input layer to the hidden layer and the hidden layer to the output layer for each encoder layer. Interestingly, despite giving MA-BERT$_{base}$ the opportunity to learn a different mapping function to approximate softmax in different encoder layers, each 2-layer neural network ended up learning a similar mapping. In fact, when we added the constraint of sharing the 2-layer neural network across all 12 encoder layers in MA-BERT$_{base}$ (Shared Softmax), a similar mapping is also learnt. Note that the mappings are rather sparse and do not seem to be random, which suggests that the 2-layer neural network has learnt something meaningful. The same observation is made when we looked into the learnt weights in MA-DistilBERT in Figure 6. In fact, we can see that the mappings learnt in MA-DistilBERT and MA-BERT$_{base}$ are similar as well. This further suggests that there is some underlying function that can approximate the role of softmax in self-attention and we believe that this will serve as an interesting point for future research.

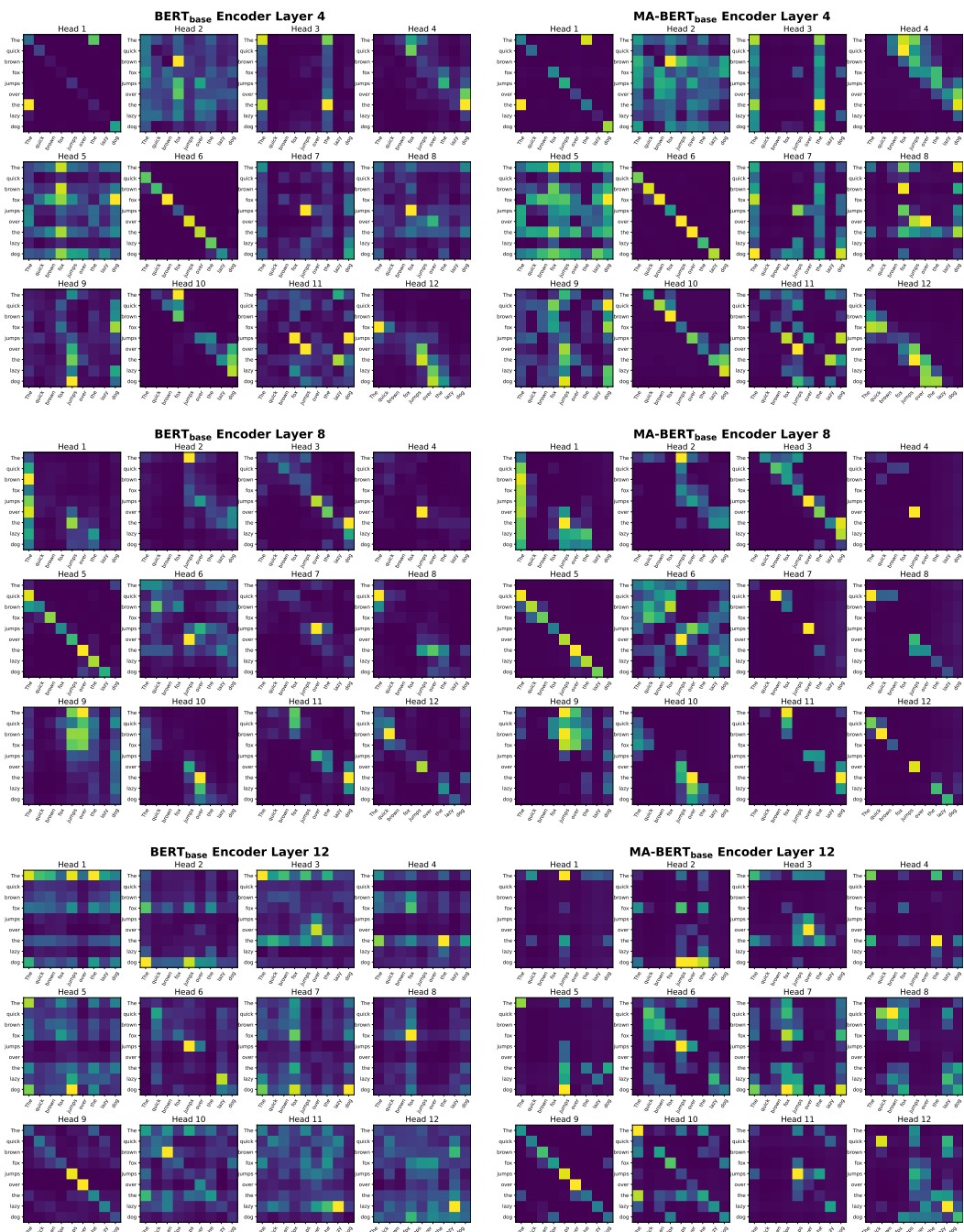

Figure 3: Attention scores produced after multi-head self-attention in encoder layer 4, 8, 12 of BERT_base (left) and MA-BERT_base (right).

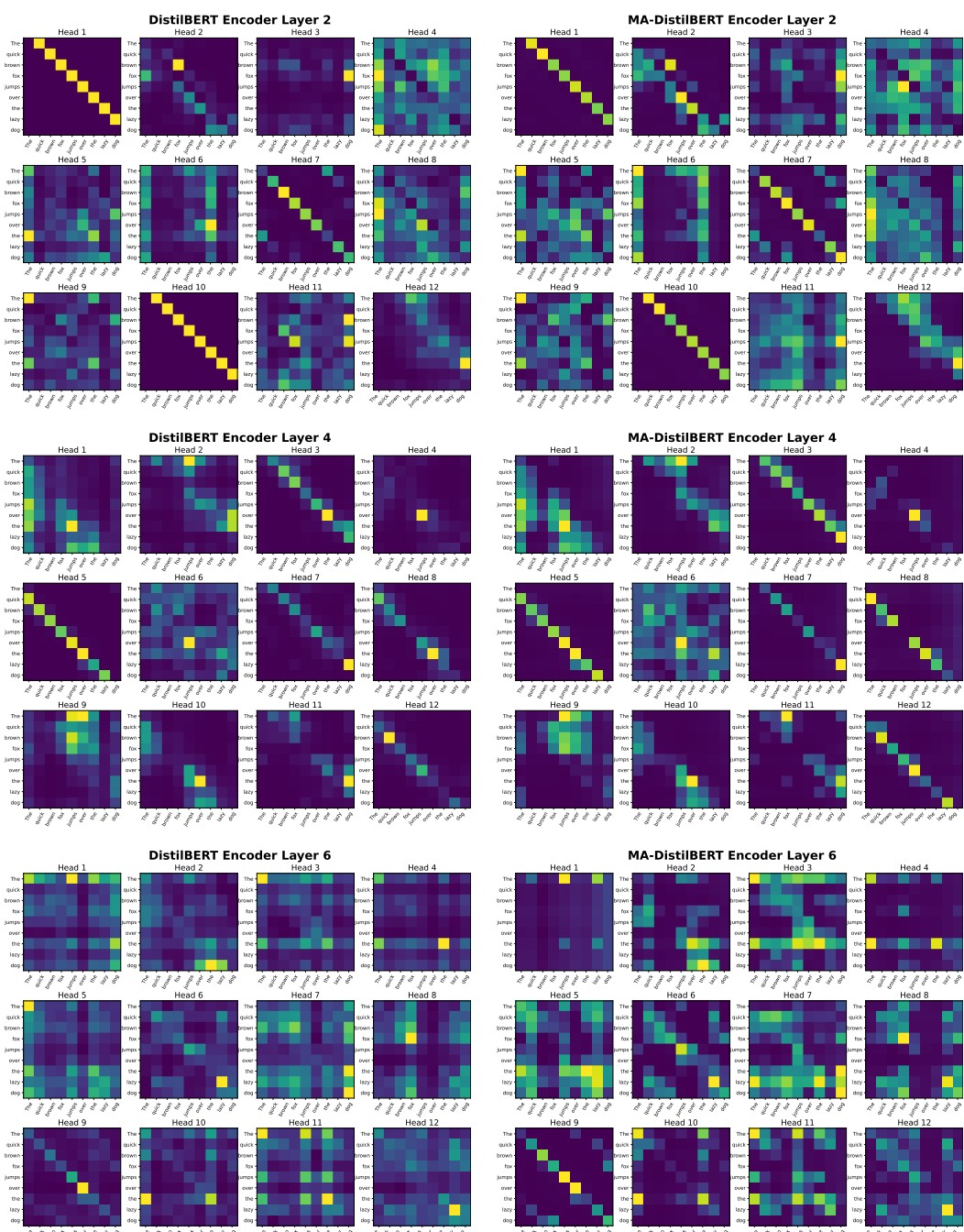

Figure 4: Attention scores produced after multi-head self-attention in encoder layer 2, 4, 6 of DistilBERT (left) and MA-DistilBERT (right).

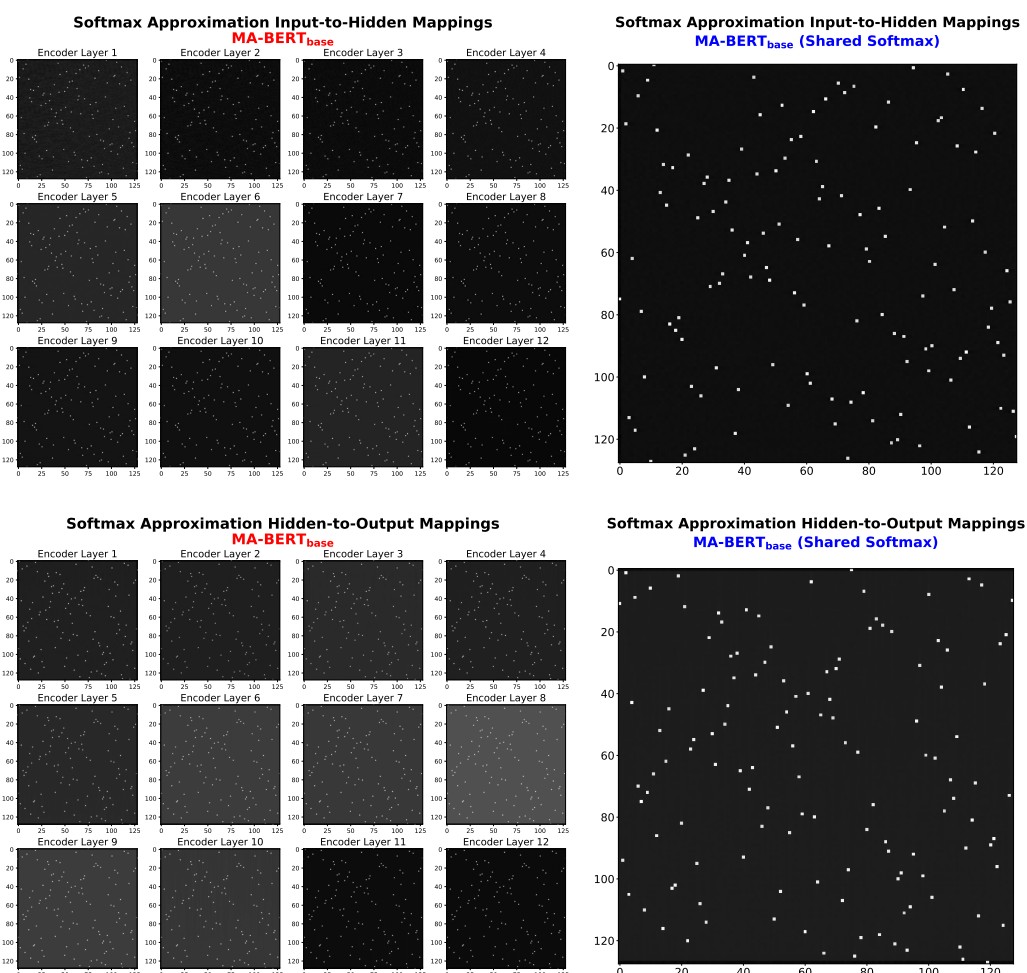

Figure 5: Input-to-hidden (top) and hidden-to-output (bottom) mappings learnt by the 2-layer neural network in each MA-BERT$_{base}$ encoder layer and the shared 2-layer neural network in MA-BERT$_{base}$ (Shared Softmax).

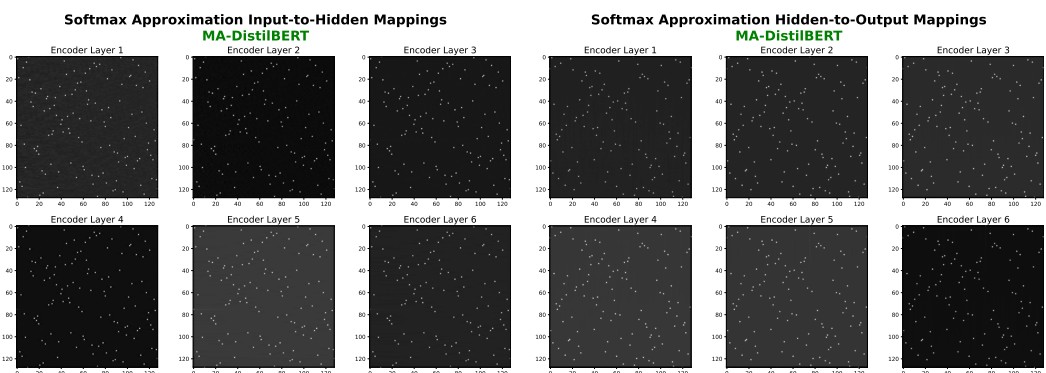

Figure 6: Input-to-hidden (left) and hidden-to-output (right) mappings learnt by the 2-layer neural network in each MA-DistilBERT encoder layer.

