# OpenReview forum: "MA-BERT: Towards Matrix Arithmetic-only BERT Inference by Eliminating Complex Non-Linear Functions"
_ICLR.cc/2023/Conference — ICLR 2023 poster_

### Official Review · Reviewer_RFc3 · 2022-10-13

**Confidence:** 4
**Correctness:** 3
**Technical Novelty And Significance:** 3
**Empirical Novelty And Significance:** 3
**Recommendation:** 5

**Clarity, Quality, Novelty And Reproducibility:**

Clarity: 5/10

Quality: 5/10

Novelty: 7/10

Reproducibility: 5/10

**Strength And Weaknesses:**

Strengths:
1. The tackled problem is relevant to the ICLR community.
2. The contributions look solid.

Weaknesses:
1. It is recommended to evaluate the proposed method executed on AI accelerators, rather than only on CPUs.
2. The reason why the knowledge transfer is needed (instead of training from scratch the MA-BERT) is unclear. Please provide more details on it.
3. The results do not seem to provide a significant improvement compared to the related works.
4. It would be useful to provide the source code for reviewers' inspection during the rebuttal.

**Summary Of The Paper:**

Transformers are complex and require several compute-intensive operations. This paper proposes to simplify the execution of the BERT inference by using matrix arithmetic-only operations. The results show that the proposed method achieves considerable inference time reduction with relatively low accuracy loss compared to the baseline.

**Summary Of The Review:**

Borderline paper where several concerns should be clarified.

---

> ### Author Response · Authors · 2022-11-17
> **Official Response to Reviewer RFc3's Comments (Part 1 of 2)**
>
> Dear Reviewer,
>
> Thank you for taking the time to review our paper. We are happy to hear that you have found our work relevant to the ICLR community and that our contributions are solid.
>
> The following are our responses to your comments on the shortcomings of our paper.
>
> **Comment 1:** *It is recommended to evaluate the proposed method executed on AI accelerators, rather than only on CPUs.*
>
> Thank you for the recommendation. In addition to CPU, we have done the inference experiment on GPU, which is the most popular hardware accelerator for deep learning, as well as on an FPGA with results shown below.
>
> For the experiment on GPU, we implemented DistilBERT and MA-DistilBERT in Cuda C++ using Thrust and cuBLAS libraries with a single NVIDIA GeForce RTX 3090 as the GPU. The following table shows the median GPU inference time per encoder layer for DistilBERT and MA-DistilBERT for 5 different runs of 1000 inferences each using the IMDb test dataset. Additionally, we present the normalized cycle count to execute a single encoder layer of MA-DistilBERT and DistilBERT on a Xilinx Virtex UltraScale+ FPGA VCU118 Evaluation Kit under the constraint of 15k LUT resources:
>
> |               | Inference on RTX 3090 GPU | Inference on Virtex UltraScale+ FPGA |
> |---------------|---------------------------|--------------------------------------|
> | DistilBERT    | 2.37 ms                   | 1.89 normalized cycles                         |
> | MA-DistilBERT | 1.39 ms                   | 1 normalized cycle                         |
>
> From our results, we can see that MA-DistilBERT is able to offer around 41% and 47% speedup on GPU and FPGA, respectively, as compared to the baseline DistilBERT.
>
> Note that BERT*base* is just DistilBERT with twice the number of encoder layers and the speedups reported are applicable to BERT*base* as well. The same goes for MA-DistilRoBERTa since its encoder layer is the same as MA-DistilBERT.
>
> **Comment 2:** *The reason why the knowledge transfer is needed (instead of training from scratch the MA-BERT) is unclear. Please provide more details on it.*
>
> Intuitively, having a good baseline model to guide a student model allows the student model to learn more easily and quickly as compared to having the student model learn from scratch without any guidance. In fact, we can also draw parallels between how effectively we, as humans, can learn with and without a teacher’s guidance.
>
> For more concrete proof, we conducted an experiment by training MA-BERT*base* from scratch, as you have suggested, and we found that MA-BERT*base* had difficulty learning - its Masked Language Modelling (MLM) loss did not decrease after 1 epoch of training and the highest MLM accuracy achieved on our validation dataset was only 4.5%. For MA-DistilBERT, while it was able to learn, the highest MLM accuracy achieved after 3 epochs of pretraining was 26.3%. In comparison, with knowledge transfer, MA-BERT*base* and MA-DistilBERT achieved a maximum MLM accuracy of 56.0% and 53.2%, respectively. Note that these accuracies are similar to what were achieved by BERT*base* and DistilBERT. Clearly, knowledge transfer helps MA-BERT to learn more easily.
>
> Moreover, in the original BERT paper, Devlin et al. (2018) mentioned that BERT*base* was pre-trained for approximately 40 epochs over the BookCorpus and Wikipedia Corpus using a total of 16 TPU chips, which took around 4 days to complete. If we were to train MA-BERT from scratch, similar hardware resources and time would be required, assuming that MA-BERT could even achieve a similar performance as BERT. This is clearly prohibitive to anyone who is conducting research with an academic budget. In many cases, only large industry players such as Google, Facebook, and Microsoft can afford to invest such resources. With knowledge transfer, MA-BERT*base* was able to achieve good performance that is on par with its teacher, BERT*base*, with just 3 epochs of pretraining using a single GPU. The same applies to MA-DistiilBERT and MA-DistilRoBERTa. Evidently, the benefits of knowledge transfer in terms of time and resource savings can be seen.

---

> > ### Comment · Reviewer_RFc3 · 2022-11-21
> > **Response to Authors**
> >
> > The efforts made by the authors in answering the reviewers' comments are appreciated. The score is confirmed.

---

> > > ### Author Response · Authors · 2022-11-22
> > > **Clarification Regarding the Given Recommendation Score**
> > >
> > > Dear Reviewer,
> > >
> > > Thank you for the reply.
> > >
> > > We noticed that you maintained the original score of 5 despite us addressing all four of the concerns mentioned in your official review. As such, can we kindly check if there is something lacking in our paper or rebuttal that is preventing you from raising the score? We would be happy to address any additional comments that you have regarding our paper.
> > >
> > > Your response will be greatly appreciated.

---

> ### Author Response · Authors · 2022-11-17
> **Official Response to Reviewer RFc3's Comments (Part 2 of 2)**
>
> **Comment 3**: *The results do not seem to provide a significant improvement compared to the related works.*
>
> In addition to the 27% speedup on CPU, we have done the inference experiment on GPU and achieved a 41% speed up, which is a significant improvement on general AI accelerator like GPU. Furthermore, our work is compatible with other optimization methods such as model quantization. In addition, having matrix arithmetic-only operations not only leads to a speed-up in inference timings but also enables much more efficient and low cost hardware design, which we believe is valuable to the ICLR community, as you have indicated, as well as the system and hardware design community. Please refer to the following for our further justification:
>
> - In our paper, we proposed several modifications to the architecture of BERT to allow a more hardware-efficient inference. To the best of our knowledge, we believe that our paper is the first to attempt to eliminate the complex non-linear functions in BERT and we have demonstrated that it is possible to achieve similar performance as the baseline with the aid of knowledge transfer.
>
> - Furthermore, we managed to approximate the workings of softmax in the self-attention layers using a 2-layer neural network. We have added additional empirical evidence in the appendix that demonstrates its success and further suggests the existence of some underlying function that can approximate what softmax is doing.  Please take a look at the new materials that we have added under Appendix A2 Analysis of Softmax Approximation. We believe that this will be an interesting point for future research.
>
> - Lastly, we believe that our work can path the way for future works which can bring MA-BERT to greater heights. One possible direction would be to quantize MA-BERT to integer precisions for even more efficient inference. Due to the complex non-linear functions in BERT, many previous works on quantizing BERT have involved well-thought-out quantization and de-quantization procedures during the transitions between linear operations and non-linear operations. This is due to the fact that non-linear operations typically require higher precision in order to achieve adequate accuracy. As such, given that all operations (except ReLU) in MA-BERT can be expressed as matrix arithmetic, we believe that quantizing MA-BERT would be less challenging and can serve as a possible direction for research.
>
> **Comment 4:** *It would be useful to provide the source code for reviewers' inspection during the rebuttal.*
>
> As you requested, the source code can be found at the following link: https://anonymous.4open.science/r/MA-BERT-1EFE/
>
> You will be able to find the code for MA-BERT*base*, MA-BERT*base* (Shared Softmax), and MA-DisitlBERT as well as their pre-trained checkpoints that we have obtained through knowledge transfer from BERT*base* and DistilBERT. We have tested and included a Jupyter Notebook to load the different models from its pre-trained checkpoint for your convenience. Unfortunately, due to copyright reasons, we are unable to release the pre-training and fine-tuning code as of now.
>
> We hope that we have addressed most, if not all, of your current concerns. Nonetheless, if you still have any other doubts about our paper, do let us know and we would be more than happy to clarify them.

---

### Official Review · Reviewer_K662 · 2022-10-25

**Confidence:** 3
**Correctness:** 3
**Technical Novelty And Significance:** 2
**Empirical Novelty And Significance:** 2
**Recommendation:** 5

**Clarity, Quality, Novelty And Reproducibility:**

Novelty: The techniques seem to be novel. However, I am not an expert in this area.

Clarity: The reasons why the approximations work is not clear. At least an intuitive explanation would have been good. Universal approx. theory based explanation is usually too sparse.

**Strength And Weaknesses:**

Strengths:

* The techniques are simple to understand

Weaknesses

* The reasons as to why you can approximate the softmax using a 2-layer NN is unclear. For example, performer had clear justification why the projection was approximating the softmax. The argument in this paper is poor (universal approx. theory)
* No hardware results, but just software CPU implementation.


**Summary Of The Paper:**

The paper introduces 3 approximations to remove expensive non-linearities in transformers using cheaper matrix math and ReLU activation functions. Next, they perform a knowledge transfer between the trained BERT model and the approximated model to bring the accuracy up to standard in MA-BERT. The results show comparable accuracy while being cheaper to implement MA-BERT in hardware.

**Summary Of The Review:**

I enjoyed the premise of the paper and Figure 1, which shows how much hardware cycles are spent in the non-linear computations. The techniques seem simple and easy to understand. However, it is unclear why they would work. The authors should explain why softmax can be approximated using a 2-layer FFN with ReLU. It is not clear to me. Citing universal approx. theory is not enough. For inspiration, look at performer or linformer papers.

I did not see ablations on the techniques suggested in the paper. Since, these approximations are not grounded in theory, I expect more ablations to show which technique mattered more and also why.

I am not an expert in this area, however has a working knowledge base in it. The paper is simple, but lacks justifications for the choices made and hence my score.

---

> ### Author Response · Authors · 2022-11-17
> **Official Response to Reviewer K662's Comments**
>
> Dear Reviewer,
>
> Thank you for taking the time to review our paper. We are glad to hear that you enjoyed our introduction and found our paper simple to understand.
>
> The following are our responses to your comments on the shortcomings of our paper.
>
> **Comment 1:** *The reasons why you can approximate the softmax using a 2-layer NN are unclear. The argument in this paper is poor (universal approx. theory). The reasons why the approximations work is not clear. At least an intuitive explanation would have been good. Universal approx. theory-based explanation is usually too sparse. The authors should explain why softmax can be approximated using a 2-layer FFN with ReLU. It is not clear to me. Citing universal approx. theory is not enough.*
>
> Firstly, we show that as the number of hidden neurons increases, the mean squared error decreases.
>
> | Number of Hidden Neurons | Mean Squared Error |
> |--------------------------|--------------------|
> | 32                       | 1.76e-4            |
> | 128                      | 9.01e-5            |
> | 512                      | 6.08e-5            |
> | 2048                     | 5.29e-5            |
> | 8192                     | 4.90e-5            |
> | 16384                    | 3.59e-5            |
>
> To further show that a 2-layer neural network can approximate the workings of softmax during self-attention, we added the following empirical evidence to the appendix of our paper:
>
> 1.  A visualization of the output of multi-head self-attention for various layers of MA-BERT*base* and BERT*base*. It can be observed that the outputs produced by MA-BERT*base* and BERT*base* after multi-head self-attention are similar. Do note that these are the outputs produced right after the 2-layer neural network in MA-BERT*base* and the softmax function in BERT*base*.
> 2.  A visualization of the learnt weights mapping the input layer to the hidden layer and the hidden layer to the output layer for the 2-layer neural networks. We found that despite giving MA-BERT*base* the opportunity to learn a different neural network for different encoder layers, it ended up learning a similar mapping to approximate softmax in each encoder layer. In fact, when we added the constraint of sharing the 2-layer neural network across all 12 encoder layers in MA-BERT*base* (Shared Softmax), we found that a similar mapping is learnt. Note that the mappings are rather sparse and do not seem to be random, which suggests that the 2-layer neural network has learnt something meaningful.
>
> The same observations mentioned in both points 1 and 2 were made for MA-DistilBERT as well.
>
> Please take a look at the new materials that we have added under Appendix A2 Analysis of Softmax Approximation and be convinced that there is perhaps some underlying function capable of approximating what the softmax function is doing in the self-attention layers. Undeniably, it might not be the exact softmax function that the 2-layer neural network is trying to learn. Rather, it could be a generalization of what the softmax function is trying to achieve and we believe that this finding could be a notable contribution to the ICLR community as well.
>
> **Comment 2:** *No hardware results, but just software CPU implementation.*
>
> Thank you for your comment. In addition to CPU, we have done the inference experiment on GPU, which is the most popular hardware accelerator for deep learning, as well as on an FPGA with results shown below.
>
> For the experiment on GPU, we implemented DistilBERT and MA-DistilBERT in Cuda C++ using Thrust and cuBLAS libraries with a single NVIDIA GeForce RTX 3090 as the GPU. The following table shows the median GPU inference time per encoder layer for DistilBERT and MA-DistilBERT for 5 different runs of 1000 inferences each using the IMDb test dataset. Additionally, we present the normalized cycle count to execute a single encoder layer of MA-DistilBERT and DistilBERT on a Xilinx Virtex UltraScale+ FPGA VCU118 Evaluation Kit under the constraint of 15k LUT resources:
>
> |               | Inference on RTX 3090 GPU | Inference on Virtex UltraScale+ FPGA |
> |---------------|---------------------------|--------------------------------------|
> | DistilBERT    | 2.37 ms                   | 1.89 normalized cycles                         |
> | MA-DistilBERT | 1.39 ms                   | 1 normalized cycle                         |
>
> From our results, we can see that MA-DistilBERT is able to offer around 41% and 47% speedup on GPU and FPGA, respectively, as compared to the baseline DistilBERT.
>
> Note that BERT*base* is just DistilBERT with twice the number of encoder layers and the speedups reported are applicable to BERT*base* as well. The same goes for MA-DistilRoBERTa since its encoder layer is the same as MA-DistilBERT.
>
> We hope that we have addressed most, if not all, of your current concerns. Nonetheless, if you still have any other doubts about our paper, do let us know and we would be more than happy to clarify them.

---

### Official Review · Reviewer_9tBM · 2022-10-25

**Confidence:** 4
**Correctness:** 3
**Technical Novelty And Significance:** 3
**Empirical Novelty And Significance:** 3
**Recommendation:** 6

**Clarity, Quality, Novelty And Reproducibility:**

This work puts together a series of previously-proposed works to reduce the complexity of BERT models. The paper is well-written, clear and easy to understand. I believe the results of this paper can be reproduced.

**Strength And Weaknesses:**

Strengths:

-- I believe this work is well-motivated. The paper targets a very important issue (i.e., reducing latency of inference) associated with large models such as BERT.

-- The detection of the source of the latency overhead during inference is insightful as shown in Fig. 1.

-- The amount of speedup during the inference on CPUs is interesting and shows the effectiveness of the proposed techniques.

-- The paper in general is well-written and easy to understand.

Weaknesses:
-- In general, the contribution of this paper is rather limited since each of those techniques have been previously-proposed.

-- It would have been great if the breakdown of cycles in Fig. 1 was shown for the sequence length of 128 to be compatible with experimental results in Table 2.

-- My major concern about this paper is the lack of comparison with prior works. First, there is no results for inference on GPUs. Second, there is no direct comparison with prior works such as Linformer? Where does the proposed method stand w.r.t. prior works? Finally, what speedup can be obtained when using MA-DistilBERT or MA-DistilRoBERTa?

-- The GELU and softmax functions can be approximated using polynomials similar to I-BERT with no impact on accuracy.



**Summary Of The Paper:**

This paper aims at reducing the complexity of BERT models to accelerate their inference by putting together a series of previously-introduced techniques. To this end, the authors propose to approximate softmax with a two-layer neural network, replace GELU with ReLU, fuse normalization layers with adjacent linear layers, and use knowledge distillation. It has been shown that the acceleration of 1.27x can be obtained on CPUs using the aforementioned four techniques while achieving a comparable accuracy performance w.r.t. the baseline model.

**Summary Of The Review:**

In general, I believe the contribution of this paper is rather limited since the main method is a combination of previous works. On the other hand, the proposed method is simple and effective (when running on CPUs). My main concern is the lack of comparison with other works. It is not clear where this works stands w.r.t. prior works that also tries to reduce the complexity of BERT models such as Linformer and I-BERT. There is also no evaluation on GPUs while it has been mentioned as one of the motivation of this work in the introduction.

---

> ### Author Response · Authors · 2022-11-17
> **Official Response to Reviewer 9tBM's Comments (Part 1 of 2)**
>
> Dear Reviewer,
>
> Thank you for taking the time to review our paper. We are happy to hear that you have found our work well-motivated and easy to understand.
>
> The following are our responses to your comments on the shortcomings of our paper.
>
> **Comment 1**: *In general, the contribution of this paper is rather limited since each of those techniques has been previously proposed.*
>
> While we acknowledge that some of the techniques proposed in our paper are inspired by previously proposed techniques, we did not just simply put them together without further development. We proposed some novel techniques and developed some existing techniques to achieve matrix arithmetic-only BERT model with negligible accuracy loss, which has never been achieved before. Please refer to the following for our detailed justification:
>
> 1.  **The use of a 2-layer neural network to approximate the softmax function in the self-attention layers**. We have added additional empirical evidence in the appendix that proves its success and further suggests the existence of some underlying function that can approximate what softmax is doing. Please take a look at the new materials that we have added under Appendix A2 Analysis of Softmax Approximation.
> 2.  **The use of purely PowerNorm to replace LayerNorm to achieve similar results with baselines**. To elaborate on this, as mentioned in the original PowerNorm paper, Shen et al. (2020) involved a layer-scale layer before applying PowerNorm. The layer-scale layer is in fact the root mean square layer normalization (RMS LayerNorm) proposed by Zhang & Sennrich (2019). As such, what Shen et al. (2020) have done was replace LayerNorm in transformers with both RMS LayerNorm and PowerNorm to achieve better performance in several NLP tasks. This is different from what we implemented as we did not include RMS LayerNorm in MA-BERT due to the fact that its inclusion would make it impossible for us to fuse the normalization layer with neighbouring linear layers during inference. Moreover, given that RMS LayerNorm was shown to yield comparable performance against LayerNorm, there exists some ambiguity on the role of PowerNorm in helping transformers achieve good performance.
> 3.  **The complete removal of all complex non-linear functions (GELU, Softmax and LayerNorm) in BERT** to achieve matrix arithmetic-only operations with trivial ReLU in MA-BERT, which can bring hardware savings. To the best of our knowledge, we believe that our paper is the first work to eliminate the complex non-linear functions in BERT and we have demonstrated that it is possible to achieve similar performance as the baseline with the aid of customized knowledge transfer in our work.
>
> **Comment 2:** *There are no results for inference on GPUs while it has been mentioned as one of the motivations of this work in the introduction.*
>
> Thank you for the comment. We have done the inference experiment on GPU with the results shown below. We implemented DistilBERT and MA-DistilBERT in Cuda C++ using Thrust and cuBLAS libraries with a single NVIDIA GeForce RTX 3090 as the GPU. The following shows the median GPU inference time per encoder layer for DistilBERT and MA-DistilBERT after 5 different runs of 1000 inferences each using the IMDb test dataset:
>
> | **Model**     | **Inference Time** |
> |---------------|--------------------|
> | DistilBERT    | 2.37 ms            |
> | MA-DistilBERT | 1.39 ms            |
>
> Based on our implementation, we see that MA-DistilBERT is able to offer around 41% speedup on GPU as compared to the baseline DistilBERT. Note that BERT*base* is just DistilBERT with twice the number of encoder layers and the speedup is applicable to BERT*base* as well. The same goes for MA-DistilRoBERTa since its encoder layer is the same as MA-DistilBERT.
>
> **Comment 3**: *There is no direct comparison with prior works such as Linformer?*
>
> The following reasons are why we were unable to make a direct comparison with Linformer:
>
> 1.  The speedup presented in their paper only applies for an input sequence length of 512 and beyond as they are focusing on general Transformers rather than BERT. Note that BERT has a maximum input sequence length of just 512 and our paper focuses on a sequence length of 128, which is the focus of efficient design research and sufficient for many NLP tasks in scenarios where energy budget is tight.
> 2.  The source code for Linformer has not been open-sourced. While there exist some GitHub repositories that attempted to implement Linformer, they are not produced by the authors of the paper. Even if we utilize these repositories, the pre-trained checkpoints of Linformer aren’t available. This means that for us to compare accuracy performance with Linformer, we will need to train Linformer from scratch, which is prohibitively expensive.

---

> > ### Comment · Reviewer_9tBM · 2022-12-06
> > **Re: Official Response to Reviewer 9tBM's Comments**
> >
> > Thank you for your point-by-point response. Since most of my comments were properly addressed, I have increased my recommendation score from 5 to 6.

---

> ### Author Response · Authors · 2022-11-17
> **Official Response to Reviewer 9tBM's Comments (Part 2 of 2)**
>
> **Comment 4**: *Where does the proposed method stand w.r.t. prior works that also try to reduce the complexity of BERT models such as Linformer and I-BERT? The GELU and softmax functions can be approximated using polynomials similar to I-BERT with no impact on accuracy.*
>
> **Comparison with Linformer**
>
> Linformer’s goal was to reduce the time and space complexity of Transformers, particularly, in its self-attention mechanism. Different from our work, the complex non-linear functions (Softmax, LayerNorm, and GeLU) still exist in Linformer’s architecture and so do the drawbacks of these non-linear functions.
>
> As illustrated in our paper, rather than viewing Linformer’s method as a competing method, we believe our method can be complementary to Linformer’s and vice-versa. Eliminating complex non-linear functions allows Linformer to have a more hardware-efficient inference while adopting Linformer’s method in MA-BERT allows us to leverage its time-saving.
>
> **Comparison with I-BERT**
>
> I-BERT’s goal was to achieve integer-only arithmetic and this was achieved via using integer-only approximation methods for nonlinear operations. In comparison, our main objective is to achieve a BERT model with purely matrix arithmetic and trivial ReLU operations, which is accomplished by eliminating all complex non-linear functions.
>
> As stated in our paper, for LayerNorm, all features of an input token need to be processed before the mean and standard deviation statistics can be computed. Similarly, for softmax during self-attention, all inputs along the sequence dimension have to be processed before the softmax of a particular input can be computed. Both drawbacks remain in I-BERT since the non-linear functions are simply replaced with integer approximations. However, in MA-BERT, PowerNorm (which uses batch statistics during inference) takes the place of LayerNorm while a 2-layer neural network replaces Softmax. This has allowed us to achieve a modified BERT architecture where the aforementioned drawbacks are no longer present.
>
> Moreover, for accelerator designs, with all operations (except ReLU) converted to matrix arithmetic, the same existing matrix arithmetic hardware can potentially be reused. On the other hand, the integer approximations of Softmax, LayerNorm, and GELU still require separate hardware.
>
> In terms of speedup, it is difficult for us to do a fair comparison with I-BERT as they have quantized all operations to integer precision while our operations are still in 32-bit floating point precision. Nonetheless, with all operations converted to matrix arithmetic, we are optimistic that we can convert them to integer precision in the future to achieve even faster inference speedup than what we have achieved currently with floating point precision. In fact, we believe that quantizing MA-BERT would be less challenging than quantizing BERT as we no longer have to consider the quantization and de-quantization procedures that were needed to transition between linear operations and non-linear operations.
>
> **Comment 5:** *Finally, what speedup can be obtained when using MA-DistilBERT or MA-DistilRoBERTa?*
>
> Kindly note that the speedup presented under section 4.7 Evaluation of Computational Efficiency refers to the speedup achieved for a single encoder layer in BERT and MA-BERT. The encoder layers in BERT*base*, DistilBERT and DistilRoBERTa are identical. The only difference is that DistilBERT and DistilRoBERTa consist of a stack of 6 encoder layers while BERT*base* has a stack of 12 encoder layers. Hence, the speedup presented is applicable to all variants of MA-BERT.
>
> We hope that we have addressed most, if not all, of your current concerns. Nonetheless, if you still have any other doubts about our paper, do let us know and we would be more than happy to clarify them.

---

### Decision · Program_Chairs · 2023-01-20

**Decision:**

Accept: poster

**Justification For Why Not Higher Score:**

The paper could have benefitted from additional / stronger baselines.

**Justification For Why Not Lower Score:**

Quoting: This paper introduces an approach to improve inference speeds in transformer architectures, by distilling a trained model into an architecture with non-linearities removed. They find performance and speed benefits for MA-BERT compared to more traditional distillation approaches, and the authors have addressed the majority of author concerns.

**Metareview: Summary, Strengths And Weaknesses:**

This paper introduces an approach to improve inference speeds in transformer architectures, by distilling a trained model into an architecture with non-linearities removed. They find performance and speed benefits for MA-BERT compared to more traditional distillation approaches, and the authors have addressed the majority of author concerns. The paper will make a worthwhile addition to ICLR. The paper could be further strengthened by a more thorough comparison of MA-BERT versus standard distillation across different distilled model sizes, and by comparing MA-BERT to quantization (as suggested by 9tBM).

**Note From Pc:**

if the above contains the word "oral" or "spotlight" please see: "oral" presentation means -> notable-top-5% and "spotlight" means -> notable-top-25%. As stated in our emails, we are disassociating presentation type from AC recommendations